# APEX: Approximate-but-exhaustive search for ultra-large combinatorial synthesis libraries

## Abstract

Make-on-demand combinatorial synthesis libraries (CSLs) like Enamine REAL have significantly enabled drug discovery efforts. However, their large size presents a challenge for virtual screening, where the goal is to identify the top compounds in a library according to a computational objective (e.g., optimizing docking score) subject to computational constraints under a limited computational budget. For current library sizes—numbering in the tens of billions of compounds—and scoring functions of interest, a routine virtual screening campaign may be limited to scoring fewer than 0.1% of the available compounds, leaving potentially many high scoring compounds undiscovered. Furthermore, as constraints (and sometimes objectives) change during the course of a virtual screening campaign, existing virtual screening algorithms typically offer little room for amortization. We propose the approximate-but-exhaustive search protocol for CSLs, or APEX. APEX utilizes a neural network surrogate that exploits the structure of CSLs in the prediction of objectives and constraints to make full enumeration on a consumer GPU possible in under a minute, allowing for exact retrieval of approximate top-$k$ sets. To demonstrate APEX's capabilities, we develop a benchmark CSL comprised of more than 10 million compounds, all of which have been annotated with their docking scores on five medically relevant targets along with physicohemical properties measured with RDKit such that, for any objective and set of constraints, the ground truth top-$k$ compounds can be identified and compared against the retrievals from any virtual screening algorithm. We show APEX's consistently strong performance both in retrieval accuracy and runtime compared to alternative methods.

## 1 Introduction

The search for novel therapeutic agents is a cornerstone of modern medicine and drug discovery. In recent years, the emergence of ultra-large combinatorial synthesis libraries (CSLs), such as the Enamine REAL library, has significantly transformed this pursuit. These libraries, containing billions or even trillions of make-on-demand compounds, offer an unprecedented opportunity to explore a vast and diverse chemical space, significantly increasing the potential for identifying novel hit compounds with desirable properties. However, the sheer scale of these libraries presents a formidable challenge to traditional approaches to virtual screening.

State-of-the-art scoring functions used in virtual screening, like docking/affinity/co-folding scores, are too computationally expensive to render an exhaustive evaluation over modern CSLs, which number in the billions, practical. A number of virtual screening approaches have been developed to identify high-scoring compounds from large compound libraries under a limited evaluation budget. These methods include heuristic algorithms (Sadybekov et al., 2022), reinforcement learning (Pedawi et al., 2023; Klarich et al., 2024; de Oliveira et al., 2024), active learning (Graff et al., 2021; Mehta et al., 2021), and approaches that utilize generative models constrained to the library (Pedawi et al., 2022; Cretu et al., 2024; Luo et al., 2024; Gao et al., 2025). However, since these algorithms effectively assess only a small fraction of the total library—usually less than 1% of available compounds—they leave the vast majority of the chemical space unexplored and potentially overlook valuable compounds. Many of the listed strategies above include a surrogate modeling component, in which a more inference efficient model (such as a neural network) is trained to approximate the oracle scoring function to enable exhaustive evaluation with the surrogate (Gentile et al., 2020; Graff et al., 2022). But this too is impeded by the size of modern CSLs, which would naively require billions of

neural network evaluations to score exhaustively with the trained surrogate. Indeed, the growing size of CSLs and the computational demands of modern scoring functions in virtual screening create a pressing need for more efficient and comprehensive approaches.

At its core, virtual screening can be framed as a search problem, where the objective is to identify the top $k$ compounds that optimize a specific scoring function while satisfying a set of program-specific constraints, namely desired physicochemical or ADMET properties like molecular weight, lipophilicity, and permeability. The ability to effectively handle constraints is particularly crucial in a virtual screening: for any given drug discovery project, the number of compounds in a screening library that violate these constraints can be orders of magnitude larger than those that satisfy them. This often complicates the workflow and can lead to the exploration of irrelevant chemical space or aggressive post-filtering.

In this work, we introduce APEX (approximate-but-exhaustive search), a new paradigm for searching ultra-large CSLs that enables fast, declarative queries. Once trained, an APEX model allows for efficient retrieval of the (approximate) top-$k$ compounds from a CSL according to a user-specified objective subject to a set of user-specified constraints. More than a virtual screening algorithm, APEX allows for low latency exploration of massive CSLs without the need for a complex, iterative workflow. The core of this capability is a neural network surrogate model that exploits the library's combinatorial structure and amortizes the computation required for repeated querying, enabling real-time search across the entire enumerated CSL with remarkable efficiency on a modern GPU.

This paper details the theoretical underpinnings of the APEX methodology and demonstrates its practical application in virtual screening.

## 2 DATA

### 2.1 COMBINATORIAL SYNTHESIS LIBRARIES

A combinatorial synthesis library (CSL) is organized into a collection of multi-component *reactions*, each of which has a fixed number of components called *R-groups* which indicate placeholders for molecular building blocks called *synthons*. Hence, each product in a CSL can be identified by its reaction and R-group assignment. Due to their combinatorial design, commercially available make-on-demand CSLs such as the Enamine REAL library span a chemical space numbering in the tens of billions of compounds today from a few hundred thousand synthons.

In this work, we designed our own open CSL as an alternative to existing proprietary CSLs for benchmarking and reproducibility purposes. We used a random sample of 1 million "lead-like" compounds from the ZINC22 database (Tingle et al., 2023) as a starting point for library construction. Our main focus here is on developing a large virtual library of valid CSL-like molecules, so we do not consider or ensure synthetic feasibility. We used the BRICS fragmentation algorithm (Degen et al., 2008), which breaks specific bonds based on defined chemical environments, to fragment each sampled molecule into two or three fragments. Each fragment is labeled with numbered pseudoatoms at the break points, with the BRICS rules determining which pseudoatom types can be joined to form a new bond. We applied the BRICS rules (as implemented in RDKit) to enumerate two- and three-component reactions that recombined these fragments into valid chemical products. This results in a set of fragmentation rules and fragments analogous to the reactions and synthons of a CSL.

Our final CSL comprises over 10B molecules and is by design evenly split between two- and three-component reactions. Additionally, we generated two smaller libraries by uniformly downsampling each reaction. These smaller libraries contain over 12M and 1M products and are fully enumerated to enable exhaustive docking and calculation of physiochemical properties. To address data leakage concerns, for all experiments in the paper, the surrogate is trained on the 1M compound CSL and evaluation is performed using either the 12M or 10B compound CSLs.

### 2.2 DOCKING SCORES AND PHYSIOCHEMICAL PROPERTIES

For benchmarking purposes, we selected five diverse protein targets: PARP1 (an enzyme), MET (a kinase), DRD2 (a GPCR), F10 (a protease), and ESR1 (a nuclear receptor). Receptor structures and binding sites were obtained from the DOCKSTRING dataset (García-Ortegón et al., 2022).

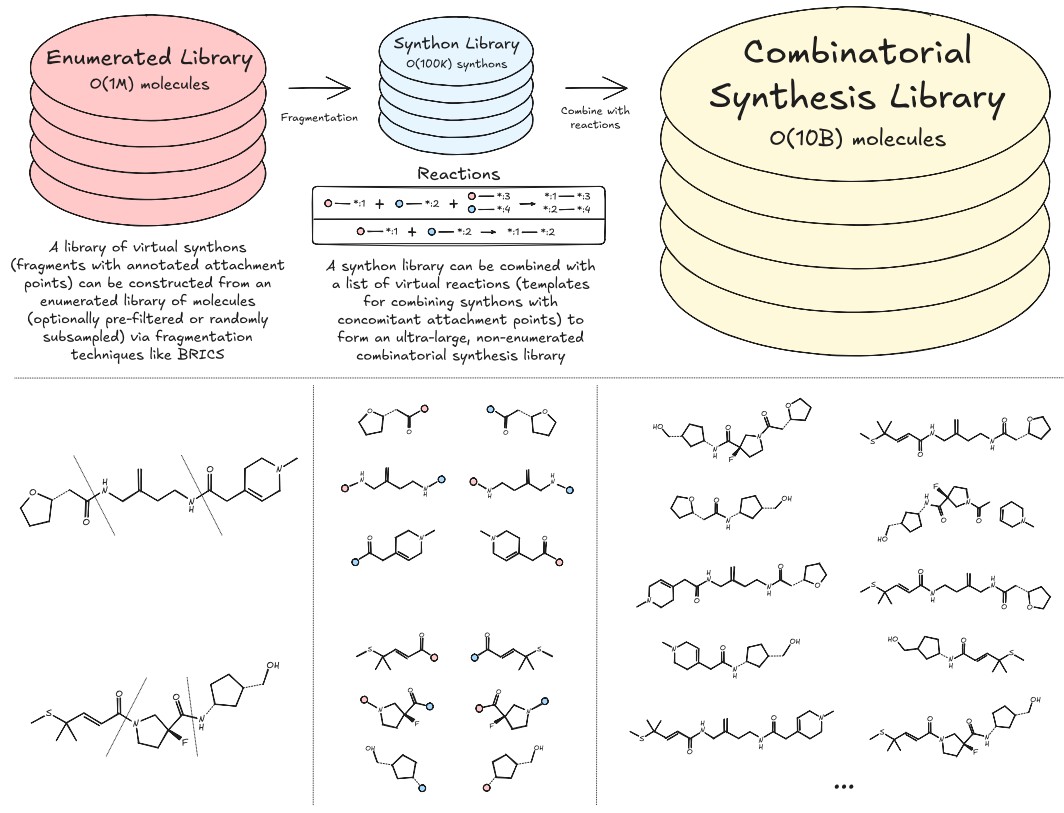

Figure 1: Beyond commercially available make-on-demand CSLs, it is relatively straightforward to design an ultra-large CSL for virtual screening using publicly available libraries of enumerated compounds like ZINC22 and cheminformatics tools like RDKit. These designs are incredibly valuable for virtual screening due to their ability to densely cover large swaths of relevant chemical space.

Molecules from this smaller library were embedded with RDKit and docked against these five targets using an accelerated implementation of the AutoDock Vina (Trott & Olson, 2010) scoring function designed to run on GPU (Morrison et al., 2020). In addition to docking scores, we calculated various physicochemical properties (e.g., molecular weight, number of hydrogen bond donors and acceptors; full list can be seen in Figure 7) for each molecule in this enumerated library.

## 3 METHODOLOGY

Given a CSL $\mathcal{D}$, which defines the chemical space $X_{\mathcal{D}}$ of eligible compounds, our goal is to identify the top-$k$ compounds from the library that maximize an objective subject to constraints. This retrieval problem can be expressed as

$$X_k^* := \arg \max_{\substack{X_k \subset X_{\mathcal{D}} \\ |X_k| \leq k}} \sum_{\mathbf{x} \in X_k} f_0(\mathbf{x}),$$

$$\text{subject to} \quad \ell_i \leq f_i(\mathbf{x}) \leq u_i, \quad \forall \mathbf{x} \in X_k, i = 1, \ldots, m, \tag{1}$$

where $f_0 : X \to \mathbb{R}$ is the objective and $f_i : X \to \mathbb{R}, i = 1, \ldots, m$, are the constraints with bounds $\ell_i < u_i$. This problem is complicated by the present and rapidly growing size of CSLs, $|X_{\mathcal{D}}| > 10^{10}$, combined with the fact that many objectives and constraints of interest—such as docking or co-folding scores—are computationally expensive to evaluate, which precludes exhaustive evaluation. We can

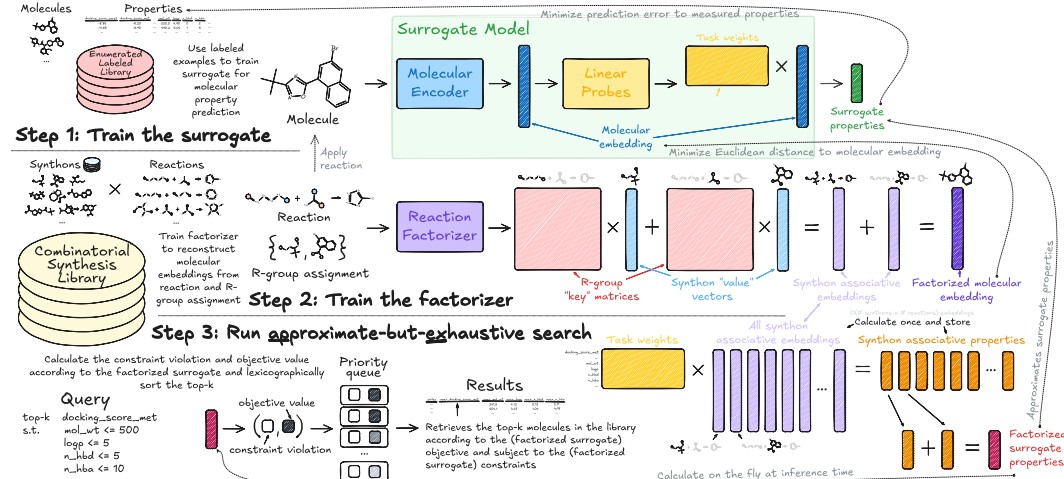

Figure 2: The APEX (approximate-but-exhaustive) search protocol, enabling rapid, on-the-fly virtual screening of ultra-large CSLs. APEX consists of three main steps. **Step 1: Train the surrogate.** Given an enumerated and labeled dataset, a multi-task neural network is trained to predict molecular properties of interest, like docking scores. **Step 2: Train the factorizer.** Given a CSL, the reaction factorizer is trained to reconstruct embeddings of the surrogate model from reaction and R-group assignment pairs. The factorizer induces an approximation of surrogate properties that is amenable to substantial amortization in executing top-$k$ retrieval on ultra-large CSLs with respect to those properties. **Step 3: Run approximate-but-exhaustive search.** Given a search query (e.g., minimize docking score on target of interest subject to drug-likeness constraints), factorized surrogate properties are calculated for all compounds in the CSL and the top-$k$ are retrieved based on the objective subject to constraints. An efficient GPU implementation allows for running a top-$k$ search with $k = 1$ million on a 10 billion compound CSL in approximately 30 seconds with a single T4 GPU.

relax (1) by substituting the original objective and constraints with surrogate models:

$$\hat{X}_k^* := \arg \max_{\substack{X_k \subset X_{\mathcal{D}} \\ |X_k| \leq k}} \sum_{\mathbf{x} \in X_k} \hat{f}_0(\mathbf{x}), \tag{2}$$

$$\text{subject to} \quad \ell_i \leq \hat{f}_i(\mathbf{x}) \leq u_i, \quad \forall \mathbf{x} \in X_k, i = 1, \ldots, m,$$

Neural network surrogates that operate directly on a 2D molecular graph or a 1D representation like SMILES are a good choice, but exhaustive evaluation of ultra-large CSLs with such surrogates is still far from a routine computational task, requiring $O(|X_{\mathcal{D}}|)$ neural network evaluations.

We develop a surrogate-based modeling strategy that permits (2) to be solved efficiently for ultra-large CSLs. First, let us discuss the parameterization of the surrogate models admissible under this design.

### 3.1 SURROGATE MODEL PARAMETERIZATION

Let $g_\theta : X \to \mathbb{R}^d$ be a neural network that encodes a molecule $\mathbf{x} \in X$ into a $d$-dimensional embedding space. We place no restrictions on $g_\theta$ beyond this, i.e., it can be a transformer that operates on the SMILES representation of $\mathbf{x}$, a GNN that operates on a 2D graph representation of $\mathbf{x}$, or some other similarly appropriate choice. We model each task $i = 0, \ldots, m$ as a linear function of the molecular embedding,

$$\hat{f}_i(\mathbf{x}) = \mathbf{w}_i^\top g_\theta(\mathbf{x}) + b_i, \tag{3}$$

where $\mathbf{w}_i \in \mathbb{R}^d$ and $b_i \in \mathbb{R}$. Given labeled data from each task, written $p_i(\mathbf{x}, y)$ where $y = f_i(\mathbf{x})$, the surrogate model is trained to minimize the prediction error relative to ground truth:

$$\min_{\theta, \{(\mathbf{w}_i, b_i)\}_{i=0}^m} \sum_{i=0}^m \mathbb{E}_{p_i(\mathbf{x}, y)} \mathbb{E}_{p(\boldsymbol{\varepsilon})} \left[ (\mathbf{w}_i^\top (g_\theta(\mathbf{x}) + \boldsymbol{\varepsilon}) + b_i - y)^2 \right]. \tag{4}$$

The surrogate is trained with noise added to the embeddings, sampled from a simple distribution $p(\boldsymbol{\varepsilon})$ like an isotropic normal. The relevance of this particular detail will be explained shortly.

## 3.2 FACTORIZATION OF SURROGATE EMBEDDINGS

As a review (see Pedawi et al. (2022) for additional details), we can represent a CSL $\mathcal{D} \equiv (T, R, S, \psi, \phi)$ hierarchically, with *synthons* (molecular fragments) $S$ at the bottom of the hierarchy, *R-groups* $R$ in the middle, and *reactions* $T$ at the top. Every synthon index $s \in S$ is associated with a corresponding molecular representation $\mathbf{x}_s \in X_*$ (again, SMILES or 2D graph), where $X_* \supset X$ extends $X$ to include attachment points, represented by the token "*". An R-group, denoted by the index $r \in R$, is comprised of a set of synthons that constitute valid assignments to the associated component in a multi-component reaction. A multi-component reaction $t \in T$, together with a valid assignment of synthons to their constituent R-groups, produces a single molecule via chemical synthesis as output. We denote by $\psi_{T \to R} : T \to \mathcal{P}(R)$ the function that returns the set of R-groups $\psi_{T \to R}(t) \subset R$ associated with a reaction $t$, where $\mathcal{P}(\cdot)$ denotes the power set function. Similarly, $\psi_{R \to S} : R \to \mathcal{P}(S)$ returns the set of synthons $\psi_{R \to S}(r) \subset S$ that can be assigned to a particular R-group. Each molecule in $\mathcal{D}$ can be referenced by a multi-index, denoted by $\boldsymbol{\chi} = (t, \{(r,s) : \exists s \in \psi_{R \to S}(r), \forall r \in \psi_{T \to R}(t)\})$, which describes the reaction and R-group assignment used to construct the molecule, $\mathbf{x} := \phi(\boldsymbol{\chi})$.

We utilize the design proposed in Pedawi et al. (2022) to model an associated hierachy of representations that describe the library at these three levels of resolution. First, the `SynthonEncoder` : $X_* \to \mathbb{R}^{d_S}$ produces an embedding for each synthon $s$ as a function of its molecular representation $\mathbf{x}_s$. Next, a deep set network called the `RgroupEncoder` : $\mathbb{R}^{d_S} \times \cdots \times \mathbb{R}^{d_S} \to \mathbb{R}^{d_R}$ produces an embedding for each R-group $r$ as a function of the representations of its constituent synthons. Finally, another deep set network, `ReactionEncoder` : $\mathbb{R}^{d_R} \times \cdots \times \mathbb{R}^{d_R} \to \mathbb{R}^{d_T}$, produces an embedding for each reaction $t$ as a function of the representations of its constituent R-groups. This is described by the following computational stack:

$$\mathbf{h}_s^S = \texttt{SynthonEncoder}(\mathbf{x}_s), \tag{5}$$

$$\mathbf{h}_r^R = \texttt{RgroupEncoder}(\{\mathbf{h}_s^S : \forall s \in \psi_{R \to S}(r)\}), \tag{6}$$

$$\mathbf{h}_t^T = \texttt{ReactionEncoder}(\{\mathbf{h}_r^R : \forall r \in \psi_{T \to R}(t)\}). \tag{7}$$

From these representations, we aim to reconstruct the molecular embedding $g_\theta(\phi(\boldsymbol{\chi}))$ as a function of the associated multi-index $\boldsymbol{\chi}$ in a manner which will permit fast and efficient approximations of (3). To do this, we model the embedding space of $g_\theta$ via a linear associative map of the R-group assignments. In particular, we introduce a `SynthonValueEncoder` : $\mathbb{R}^{d_S} \to \mathbb{R}^{d_U}$ and `RgroupKeyEncoder` : $\mathbb{R}^{d_R} \times \mathbb{R}^{d_T} \to \mathbb{R}^{d \times d_U}$ which produce intermediate representations that are combined as follows to arrive at a prediction of the molecule's latent representation:

$$\mathbf{v}_s = \texttt{SynthonValueEncoder}(\mathbf{h}_s^S), \tag{8}$$

$$\mathbf{K}_r = \texttt{RgroupKeyEncoder}(\mathbf{h}_r^R, \mathbf{h}_{\psi_{R \to T}(r)}^T), \tag{9}$$

$$\mathbf{u}_{r,s} = \mathbf{K}_r \mathbf{v}_s, \tag{10}$$

$$\hat{g}_\lambda(\boldsymbol{\chi}) = \sum_{(r,s) \in \boldsymbol{\chi}} \mathbf{u}_{r,s}. \tag{11}$$

The `SynthonEncoder`, `RgroupEncoder`, `ReactionEncoder`, `SynthonValueEncoder`, and `RgroupKeyEncoder` all combine to form the `ReactionFactorizer` or just the "factorizer" for short, which we represent by the function $\hat{g}_\lambda(\boldsymbol{\chi})$. Given a library $\mathcal{D}$ and the frozen surrogate encoder $g_\theta$, we train the factorizer to minimize the reconstruction error of the surrogate embeddings,

$$\min_\lambda \quad \mathbb{E}_{p(\boldsymbol{\chi}|\mathcal{D})} \left[ \|g_\theta(\phi(\boldsymbol{\chi})) - \hat{g}_\lambda(\boldsymbol{\chi})\|_2^2 \right]. \tag{12}$$

### 3.3 PUTTING IT TOGETHER

We can factorize the surrogate predictions by substituting (11) into (3), which simplifies as follows:

$$\hat{\hat{f}}_i(\boldsymbol{\chi}) = \mathbf{w}_i^\top \hat{g}_\lambda(\boldsymbol{\chi}) + b_i, \tag{13}$$

$$= \mathbf{w}_i^\top \left( \sum_{(r,s)\in\boldsymbol{\chi}} \mathbf{u}_{r,s} \right) + b_i, \tag{14}$$

$$= \sum_{(r,s)\in\boldsymbol{\chi}} \mathbf{w}_i^\top \mathbf{u}_{r,s} + b_i, \tag{15}$$

$$= \sum_{(r,s)\in\boldsymbol{\chi}} v_{i,r,s} + b_i, \tag{16}$$

where the $v_{i,r,s}$ terms are called *synthon associative contributions*. We use the shorthand $\hat{\hat{f}}_i(\mathbf{x})$ to denote $\hat{\hat{f}}_i(\phi(\boldsymbol{\chi}))$ when $\mathbf{x} = \phi(\boldsymbol{\chi})$, i.e., we can express $\hat{\hat{f}}_i : X_\mathcal{D} \to \mathbb{R}$. We call the expression in (16) the *approximate-but-exhaustive (APEX) factorization*, because it permits us to solve the top-$k$ problem (2) under the approximation (16) via exhaustive evaluation on $\mathcal{D}$:

$$\hat{\hat{X}}_k^* := \arg \max_{\substack{X_k \subset X_\mathcal{D} \\ |X_k| \leq k}} \sum_{\mathbf{x}\in X_k} \hat{\hat{f}}_0(\mathbf{x}), \tag{17}$$

$$\text{subject to} \quad \ell_i \leq \hat{\hat{f}}_i(\mathbf{x}) \leq u_i, \quad \forall \mathbf{x} \in X_k, i = 1, \ldots, m.$$

Since the surrogate is trained with noise added to the embeddings as per (4) (and therefore learns embeddings whose linear projections are robust to such perturbations), the APEX prediction induced by the substitution in (13) is robust to the so-called errors-in-variables problem (Griliches, 1974). The addition of isotropic normal noise in (4) is therefore a technique to statistically regularize the surrogate to ensure that it remains robust to the subsequent factorization.

To demonstrate, let's consider a simplified CSL comprised of a single three-component reaction with 10,000 distinct synthons for each R-group, i.e., $|S| = 30,000$. This yields a total of one trillion products in $\mathcal{D}$. Exhaustive screening with $\hat{f}_\theta$ would therefore require one trillion neural network evaluations. APEX, on the other hand, produces all intermediate representations for the library with just 30,000 neural network evaluations. The associative embeddings (10) can be cached as a $|S| \times d$ matrix for later re-use. Supposing $d = 1024$, this would require about 120 MB of memory. In contrast, to cache the latent representations for all of the one trillion products in $\mathcal{D}$ would require about 4 PB of memory. With the associative embeddings in our possession, we can calculate their dot products with the task weight $\mathbf{w}_i$, which is just $2|S| \times d - |S| = 61.41$ million floating point operations.

Once these terms have been computed, each $\hat{\hat{f}}_i(\mathbf{x})$ can be calculated with just a few floating point operations (three in this case: the summation of the three synthon associative contributions and the bias term $b_i$). Hence, we can approximate the surrogate predictions for all compounds in $\mathcal{D}$ with just three trillion floating point operations (i.e., 3 TFLOP). Noting that the NVIDIA Tesla T4 GPU is able to perform 8.1 TFLOPS, the APEX factorization (16) theoretically permits evaluation of all one trillion compounds in the CSL in just a few seconds.

To construct the top-$k$ set $\hat{\hat{X}}_k^*$ for the retrieval problem (17), we can stream the pre-computed synthon associative contributions for each task and add them with the bias to form the APEX prediction (16). We can compute the constraint violation under the APEX predictions,

$$\hat{\hat{c}}(\mathbf{x}) = -\sum_{i=1}^m \max(0, \ell_i - \hat{\hat{f}}_i(\mathbf{x})) - \sum_{i=1}^m \max(0, \hat{\hat{f}}_i(\mathbf{x}) - u_i), \tag{18}$$

which is zero if all of the predicted constraints are satisfied and negative if there is any violation.

Hence, for each compound in $\mathcal{D}$, we form a two-dimensional vector $(\hat{\hat{c}}(\mathbf{x}), \hat{\hat{f}}_0(\mathbf{x}))$ that is used to enter compounds into a priority queue of size $k$ that organizes them in lexicographical order. Once we have exhausted through all compounds in $\mathcal{D}$, we can remove any compound from the top-$k$ set where $\hat{\hat{c}}(\mathbf{x}) < 0$. The result is the solution $\hat{\hat{X}}_k^*$ to (17).

## 3.4 TOP-$k$ RETRIEVAL

The exposition in the previous subsection on runtime only considers evaluation of APEX predictions over the entire CSL and ignores overhead introduced by maintenance of the top-$k$ set. APEX implements custom top-$k$ algorithms for the CPU and GPU within a PyTorch CUDA C++ extension module. The CPU PyTorch operator calculates each molecule's APEX objective $\hat{\hat{f}}_0(\mathbf{x})$ and constraint violation $\hat{\hat{c}}(\mathbf{x})$ on the fly and streams them directly into a priority queue. However, APEX is uniquely suited for the GPU, as it requires only a small initial data transfer from CPU to GPU, and all intermediate calculations can be performed entirely on-GPU. To leverage the high compute capability and memory bandwidth of the GPU, the CUDA PyTorch operator employs a chain-of-batches strategy with the GPU-compatible AIR top-$k$ algorithm (Zhang et al., 2023). Additional details are provided in Appendix A.3.

## 4 EVALUATION

To demonstrate APEX's capabilities on a variety of pertinent virtual screening queries, we evaluate its ability to accurately retrieve the top-$k$ compounds in a large, representative CSL by docking score across the five selected targets (PARP1, MET, DRD2, F10, and ESR1) and against a number of relevant constraint sets used in drug discovery (described in Appendix A.4).

In all reported experiments, the surrogate is trained on the 1M compound CSL described in Section 2 (this is the only step in which labels are provided to the model) and the factorizer is trained on either the 12M or 10B compound CSL (in the absence of labels) to reconstruct embeddings produced by the trained surrogate model. We use an embedding dimension of $d = 64$. No extensive hyperparameter tuning was performed; we opted for a lightweight model for purposes of demonstrating APEX (but note that runtime for APEX search is not a function of $d$ once pre-calculations have been performed).

### 4.1 TOP-$k$ RETRIEVAL

For a library $\mathcal{D}$, objective $f_0$, constraints $\{(f_i, \ell_i, u_i)\}_{i=1}^m$, and evaluation budget $k$, we are ultimately interested in a screening algorithm's ability to accurately retrieve the ground truth top-$j$ set $X_j^*$ in (1), where $j \leq k$. For example, we might have the budget to evaluate $k = 100{,}000$ compounds but wish to quantify what percent of the top-$j = 100$ were correctly retrieved. For APEX, this quantity can be expressed simply as

$$\text{Recall-}j\text{-at-}k = \frac{|X_j^* \cap \hat{X}_k^*|}{|X_j^*|}. \tag{19}$$

Of course, this requires knowing the ground truth top-$j$ set $X_j^*$ for a given search query. We use the 12M enumerated and exhaustively scored CSL to perform such an evaluation.

Results are shown in Figure 3A. With a budget of $k = 100{,}000$ retrievals (representing 0.803% of compounds in the CSL), the ground truth top-$j$ compounds are recovered at rates far exceeding selection with a random baseline across all targets and for all sets of constraints. In addition to the search without constraints, the recall is highest for the Veber set of constraints, which are the least stringent and are satisfied by most compounds in the library (Figure 3B). The Astex Rule-of-3 constraints are designed for fragment-based drug discovery but we include them here as an example of a more stringent constraint set. While the rate of constraint satisfaction is lowest for this set, it is still much higher than the baseline rate of constraint satisfaction in the library.

### 4.2 COMPARISON WITH THOMPSON SAMPLING

As a more challenging baseline, we also compare APEX to Thompson sampling (TS). As TS is run on each reaction separately, we limit the comparison to the top five largest reactions in our 12M CSL (in total comprising over 4 million products). The total number of evaluations for TS is $|S| \times w + i$, where $S$ is the set of synthons for that particular reaction, $w$ is the number of warmup steps, and $i$ is the number of TS iterations and output molecules. We run TS for 100, 1000, or 10,000 iterations, with 3 warmup steps for two-component reactions and 10 for three-component reactions (as suggested in Klarich et al. (2024)). As this TS implementation does not directly support

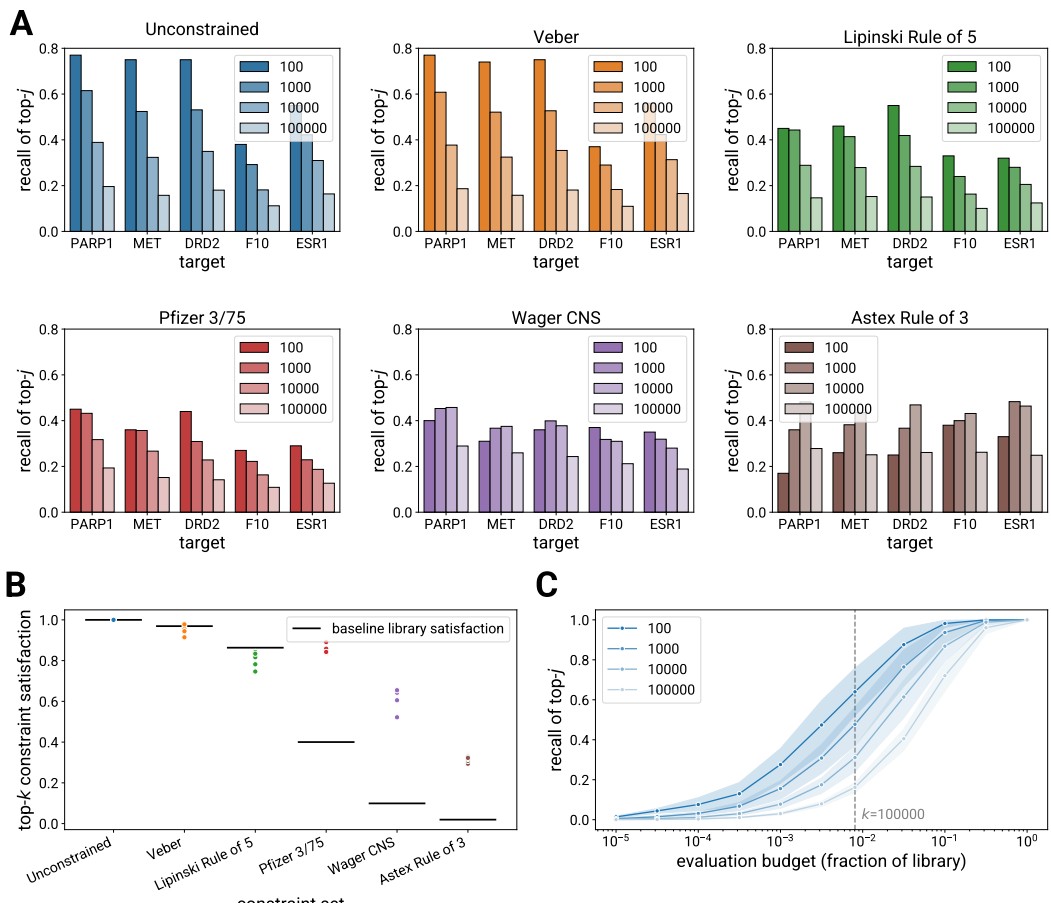

Figure 3: (A) Percent of compounds in the ground truth top-$j$ set retrieved by the APEX top $k = 100,000$ set from the 12M compound CSL. A random baseline will achieve a recall below 0.01. (B) Constraint satisfaction rates for the APEX retrievals. Black line denotes the base fraction of satisfying compounds in the library for each set of constraints. (C) Recall of different top-$j$ sets without constraints as a function of increasing evaluation budget. Recall is averaged over all five targets, with error bars showing the standard deviation. Per-target recall curves are shown in Figure 9 of the Appendix. The dashed line corresponds to $k = 100,000$, the budget used for (A) and (B).

constraints on molecular properties, we perform the comparison in the unconstrained case for both APEX and TS, only minimizing docking score as the objective. For each reaction and number of TS iterations, we set $k$ for APEX to the total number of TS evaluations, and evaluate top-$j$ recall within a particular reaction. Full results are shown in Figure 8 in the Appendix. While results vary across targets and reactions, APEX consistently outperforms or matches TS at recalling the top-$j$ compounds, showing particular strength at lower evaluation budgets.

## 4.3 DOCKING SCORE ENRICHMENT ON ULTRA-LARGE LIBRARIES

Figure 4 plots the empirical CDF of docking scores across the five targets for the APEX top-$k$ set in both the 10B and 12M compound CSLs against the background distribution of scores from the 12M compound library. This result demonstrates clear enrichment in the APEX top-$k$ sets relative to the background set, and further highlights the value of screening larger CSLs to identify higher scoring compounds enabled by APEX's accelerated runtime and ability to scale to ultra-large combinatorial libraries.

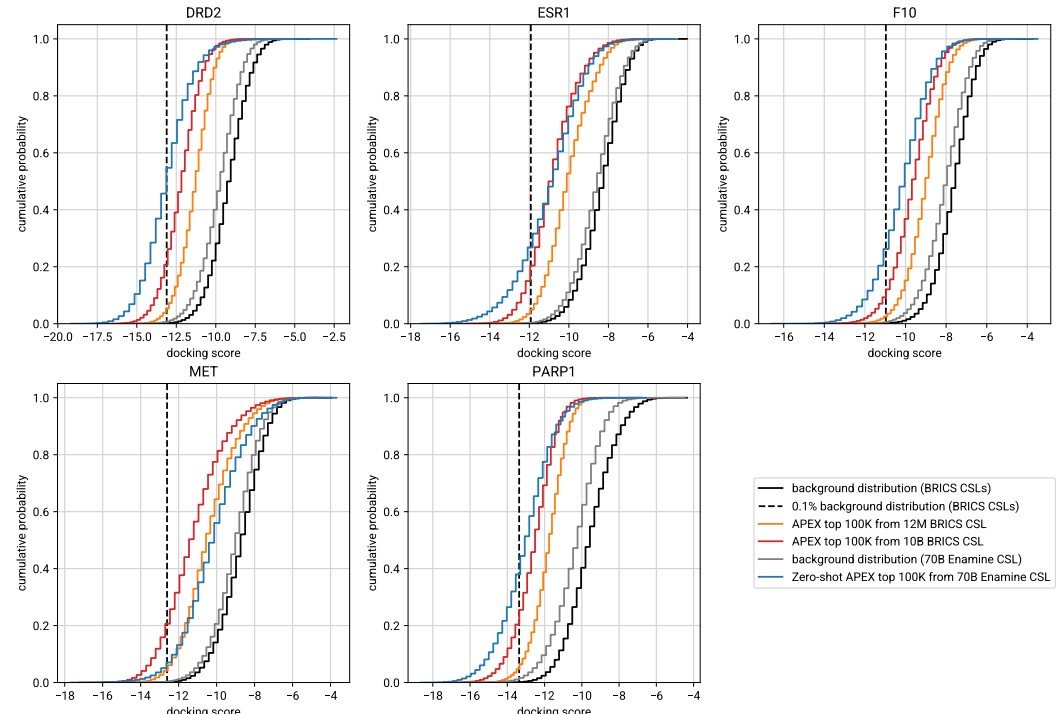

Figure 4: Docking scores for the APEX top $k = 100{,}000$ on the 10B library are enriched with respect to the background distribution and with respect to the top-$k$ set from the smaller 12M library. Lower scores are better (i.e., indicate better interaction between ligand and receptor).

### 4.3.1 ZERO-SHOT APPLICATION TO THE ENAMINE REAL LIBRARY

In addition to the BRICS CSLs, we also apply APEX to the commercial Enamine REAL library (9-2024 version). This library contains more than 70B compounds and serves as a test of APEX's generality, both in scaling to even larger library sizes and as an application of a pretrained surrogate and factorizer in a zero-shot manner.

Figure 4 presents the docking score distributions from this library alongside a background score distribution generated from 100,000 random compounds. Despite the surrogate and factorizer being trained on a different, much smaller library, APEX is able to enrich docking scores with respect to the background distribution of the Enamine library and, in most cases, with respect to the top-$k$ of the 10B BRICS library. The lowest enrichment is from MET kinase, which also corresponds to the largest drop-off in R-squared in this zero-shot application of the factorizer (Figure 7 in the Appendix). While these results demonstrate APEX's capabilities in a zero-shot context on ultra-large vendor CSLs today, even greater docking score enrichment is likely achievable through fine-tuning the surrogate (and subsequently the factorizer) using labeled data from the target CSL.

### 4.4 EXECUTION SPEED OF APEX ON ULTRA-LARGE CSLS

Table 1 reports runtimes of APEX top-$k$ search on the BRICS and Enamine libraries for different choices of $k$, evaluated on a single NVIDIA Tesla T4 GPU. The reported runtimes represent end-to-end execution, i.e., from problem specification to an output dataframe with APEX top-$k$ SMILES and their associated APEX-predicted objective and constraint values. In screening the 10B and 70B compound libraries, we observe an order of magnitude speedup in runtime when using the GPU top-$k$ implementation as opposed to CPU. Further, as constraints are included, the gap widens significantly, with the CPU implementation's runtime increasing approximately linearly in the number of constraints added. Using the GPU top-$k$ implementation, APEX is able to retrieve the approximate top $k = 1{,}000{,}000$ compounds from a 10B compound library in less than thirty seconds under

standard drug likeness constraints, making it a highly performant and scalable search protocol for ultra-large CSLs.

| | Unconstrained | | | Lipinski Rule of 5 | | |
|---|---|---|---|---|---|---|
| $k$ | BRICS 12M | BRICS 10B | REAL 70B | BRICS 12M | BRICS 10B | REAL 70B |
| 10,000 | 0.3 (0.4) | 10.9 (130.7) | 168.4 (838.5) | 0.3 (0.7) | 13.9 (437.7) | 186.1 (3163.2) |
| 100,000 | 1.2 (1.2) | 11.6 (131.4) | 169.3 (847.2) | 0.9 (1.7) | 14.6 (443.7) | 187.5 (3184.6) |
| 1,000,000 | 10.9 (12.7) | 21.2 (147.6) | 184.0 (858.9) | 10.8 (12.9) | 24.3 (462.2) | 202.4 (3142.5) |

Table 1: Runtime of APEX top-$k$ search across constraints and library sizes in seconds. Times are averaged over five runs (one with each target's docking score as an objective), with GPU runtime reported first and CPU runtime reported in parentheses.

## 5 CONCLUSION

In this paper, we proposed the APEX search protocol for the virtual screening of combinatorial synthesis libraries, enabling the rapid execution of declarative queries that scales to ultra-large libraries (in excess of 10 billion compounds). While traditional virtual screening algorithms are limited by design to evaluate only a small fraction of the eligible search space, APEX enables a fast, exhaustive evaluation over the entire search space by taking advantage of the structure of CSLs. This allows researchers to rapidly identify high-scoring compounds virtually that satisfy design constraints. We demonstrated APEX's capabilities on a benchmark CSL of over 10 million compounds, all annotated with ground truth docking scores and physiochemical properties. Our results show that APEX consistently achieves high recall rates for the ground truth top-$k$ compounds at low $k$ and effectively satisfies diverse constraint sets, far exceeding random baselines.

APEX is a significant step towards making exhaustive virtual screening a routine computational task. Its ability to efficiently screen entire CSLs ensures that valuable, high-scoring compounds are not overlooked. Moreover, due to its rapid execution speed—virtually screening a CSL in excess of 10 billion compounds in less than 30 seconds with a single Tesla T4 GPU—APEX enables rapid hypothesis testing and interactive exploration of chemical space.

## USE OF LARGE LANGUAGE MODELS (LLMS)

The use of LLMs in this paper was limited to minor stylistic and grammatical improvements.

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

# A APPENDIX

## A.1 THE VIRTUAL LIBRARY

Figure 5 displays twenty randomly selected molecules from the 10B compound CSL constructed as part of this study. In Figure 6, the distribution of molecular properties for the fully enumerated 12M compound CSL are shown for both two- and three-component reactions. We note that compounds originating from three-component reactions tend to be larger than those from two-component reactions.

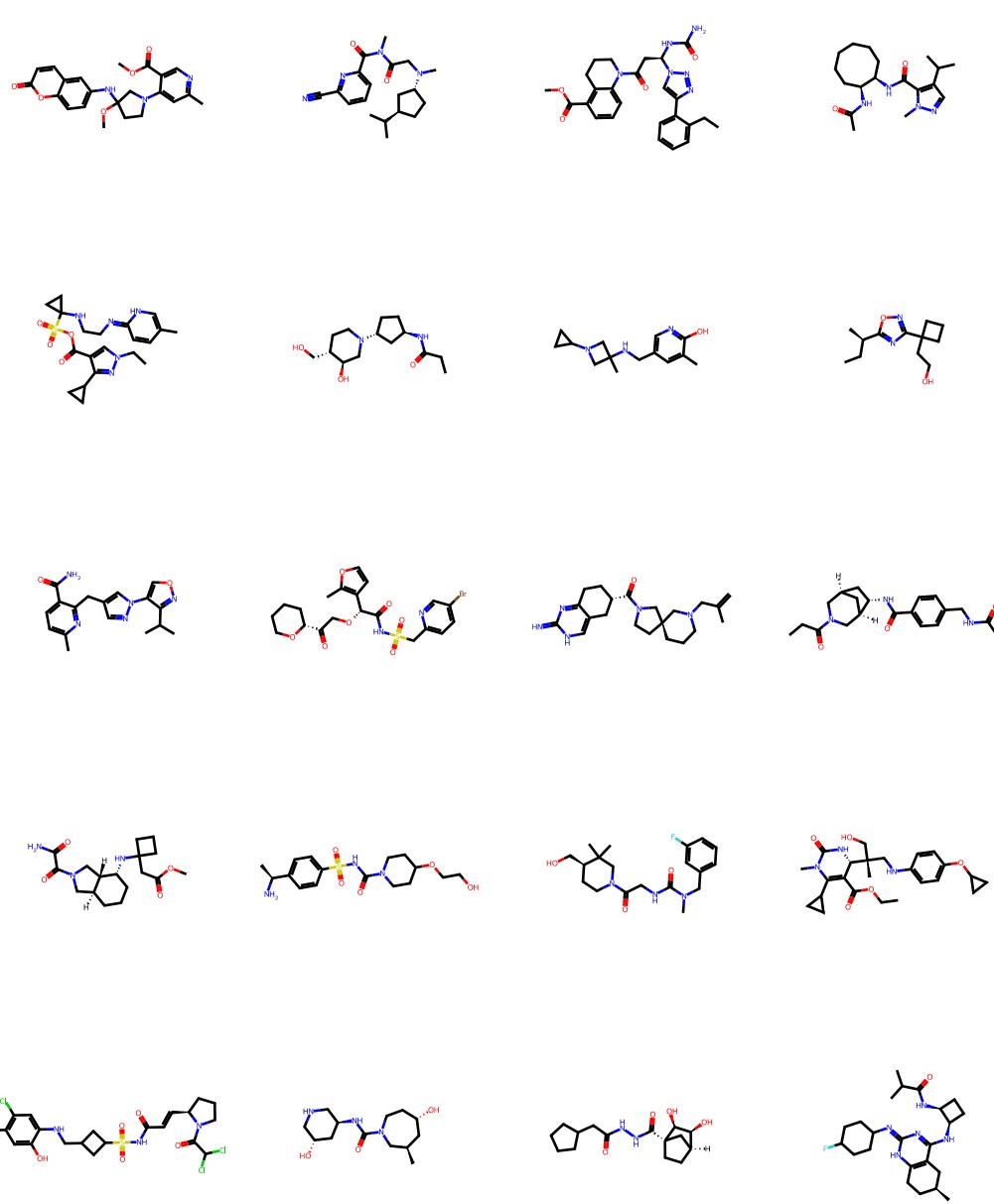

Figure 5: Example molecules from the 10B compound CSL.

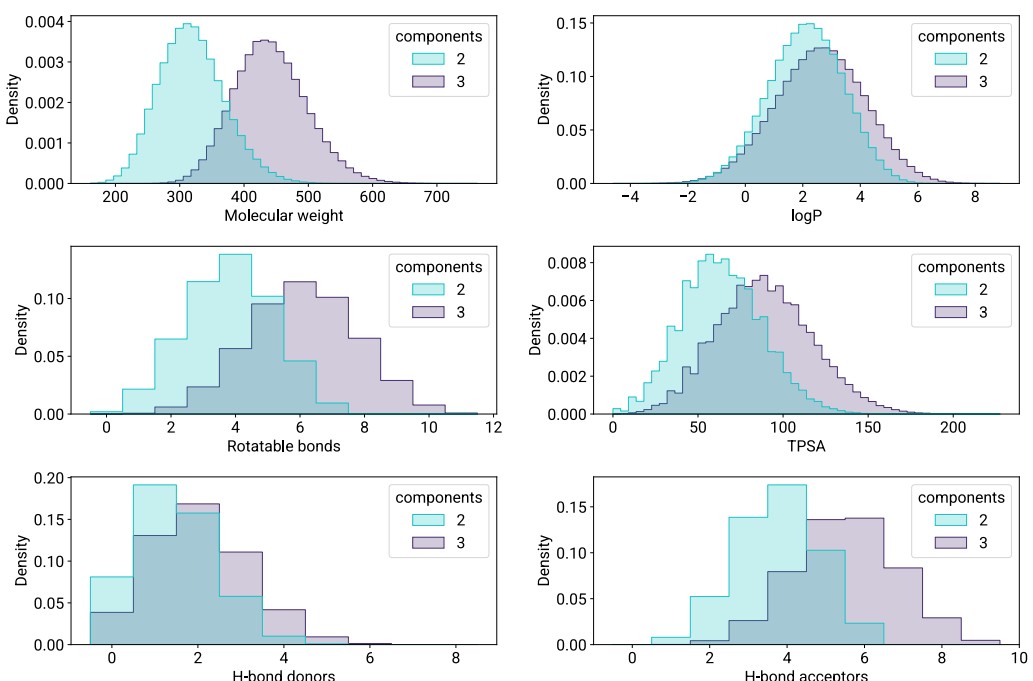

Figure 6: Distribution of molecular properties in the 12M compound CSL.

## A.2 REGRESSION PERFORMANCE BY ENDPOINT

Figure 7 displays the R-squared for the surrogate (original and APEX-factorized) across all 28 endpoints considered in this study measured on a random sample of compounds from the 12M fully enumerated CSL. For the five docking score endpoints we additionally report rank correlation with Kendall's tau-b and Spearman's rho.

## A.3 GPU IMPLEMENTATION OF FACTORIZED TOP-$k$ SEARCH

The factorized top-$k$ search employed in APEX is particularly well suited for GPUs. Each operation (score calculation, element tracing, and index decoding) can be performed independently for each compound in the CSL. Moreover, NVIDIA's CCCL library (CCCL Development Team, 2023) provides an efficient batch-based AIR top-$k$ method (Zhang et al., 2023), which we leverage in our implementation using a chain-of-batches strategy.

We first partition the CSL into batches of (reaction, first R-group assignment) pairs of some chunk size (e.g., one billion compounds) and evaluate scores, the two-dimensional pair $(\hat{\hat{c}}(\mathbf{x}), \hat{\hat{f}}_0(\mathbf{x}))$ denoting the APEX-predicted constraint violation and objective value, for all compounds in a batch on the GPU. For example, a batch can contain all compounds from the first three reactions (all R-groups fully enumerated) and all compounds from the fourth reaction where the first R-group assigned one of the first five eligible synthons, such that the total number of products is less than or equal to the specified chunk size.

Within a batch, compound scores are computed in parallel: CUDA blocks iterate over (reaction, first R-group assignment) pairs, while threads loop through subsequent R-group assignments. Synthon associative contributions are accumulated in shared memory for higher compute throughput. If a reaction has more than two R-groups, the remaining ones are processed with plain loops.

After score computation, results are passed to CCCL's AIR top-$k$ method to filter for the top-$k$ indices for that batch. For subsequent batches, previously selected elements are prepended to the score arrays before the next AIR top-$k$ call, and the indices within the full CSL are tracked, enabling chain-of-batches. We carefully trace the movement of elements: an index larger than $k$ means a new

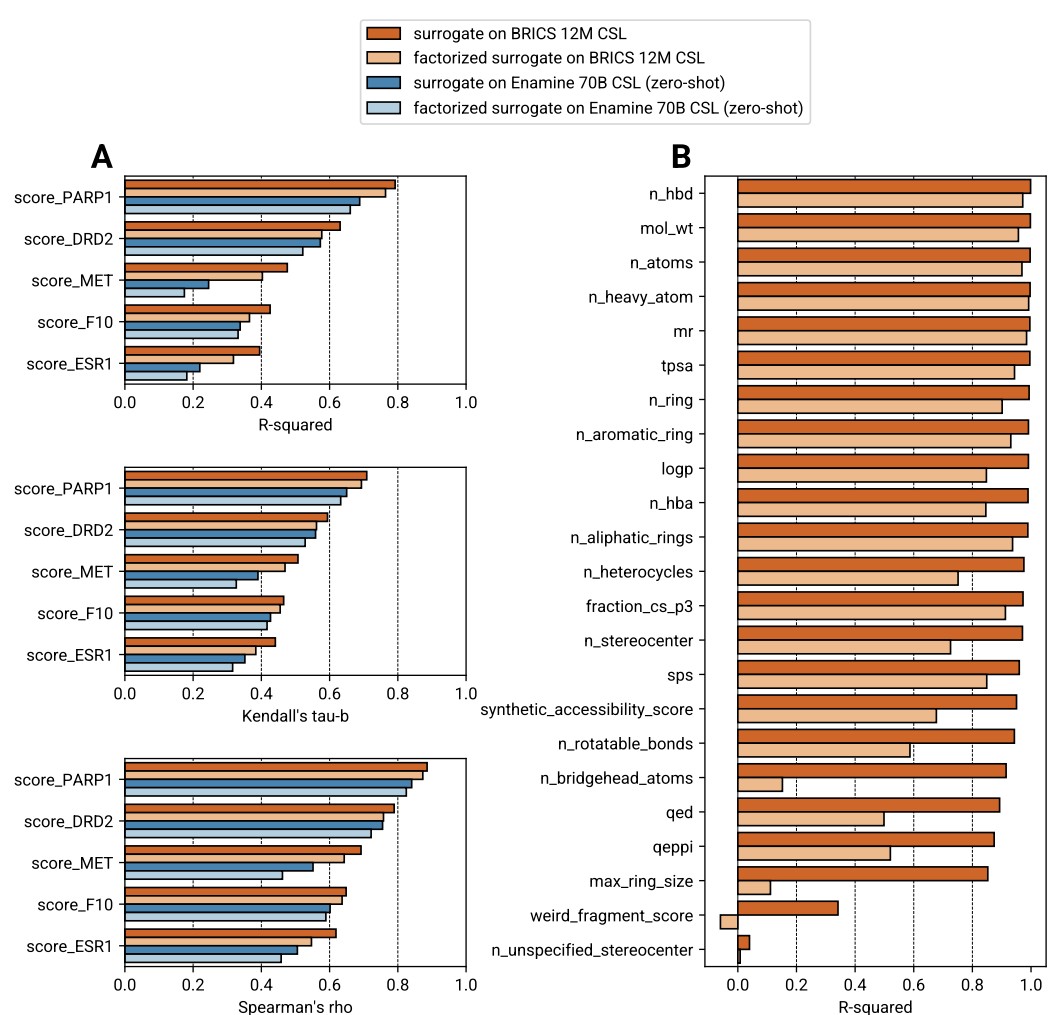

Figure 7: Accuracy of predicted (A) docking scores and (B) physiochemical properties for the original surrogate model as well as factorized version.

element from the current batch is within top $k$; otherwise the element is from previous batches but its location within top $k$ could have shifted. The kept indices array is updated accordingly.

Once the CSL is exhausted, each of the global top-$k$ indices within the library is decoded, again on GPU, using the (reaction, R-group assignment) mapping. The final results are then returned to the user for downstream processing (e.g., conversion of reaction and R-group assignment to SMILES).

## A.4 CONSTRAINT SETS

Table 2 provides details on the constraints used in this paper's experiments.

## A.5 COMPARISON WITH THOMPSON SAMPLING

Figure 8 plots the recall of APEX against Thompson sampling across the five most prevalent reactions in the 12M compound CSL and against the five targets considered in this paper.

| Rule | Property | Value |
|------|----------|-------|
| **Lipinski Rule of 5** (Lipinski et al., 1997) | Molecular weight | $\leq 500$ Da |
| | logP | $\leq 5$ |
| | H-bond donors | $\leq 5$ |
| | H-bond acceptors | $\leq 10$ |
| **Veber** (Veber et al., 2002) | Rotatable bonds | $\leq 10$ |
| | TPSA | $\leq 140$ Å$^2$ |
| **Pfizer 3/75** (Hughes et al., 2008) | logP | $\leq 3$ |
| | TPSA | $\geq 75$ Å$^2$ |
| **Wager CNS** (Wager et al., 2010) | Molecular weight | $\leq 360$ Da |
| | logP | $\leq 3$ |
| | TPSA | $\geq 40$ Å$^2$, $\leq 90$ Å$^2$ |
| | H-bond donors | $\leq 1$ |
| **Astex Rule of 3** (Congreve et al., 2003) | Molecular weight | $\leq 300$ Da |
| | logP | $\leq 3$ |
| | H-bond donors | $\leq 3$ |
| | H-bond acceptors | $\leq 3$ |
| | Rotatable bonds | $\leq 3$ |
| | TPSA | $\leq 60$ Å$^2$ |

Table 2: Constraint sets used for experiments in Figure 3.

### A.6 RECALL OF TOP-$j$ COMPOUNDS AT INCREASING EVALUATION BUDGET

Figure 9 shows the recall of top compounds (in the absence of constraints) as a function of increasing evaluation budget, expressed as the fraction of the library evaluated with the oracle.

### A.7 SCORE-BASED CONSTRAINTS AND COMPOSITE OBJECTIVES

To further test the robustness of the surrogate docking score predictions, we ran APEX search in a counter-screening scenario, where one target is chosen as the objective to minimize and constraints are added that the other four targets all score above their 50th percentile. Figure 10 shows the mean docking scores of the best 100 compounds (re-ranked by the true objective) after a $k = 100,000$ search (BRICS 12M library), represented in terms of their eCDF. While these counter-screening constraints are generally effective at increasing the "specificity" of the top compounds, they do result in worse absolute docking scores for the objective. We also tested defining a composite objective as the sum of all five targets' docking scores, which proved quite effective at finding compounds that score well across all targets.

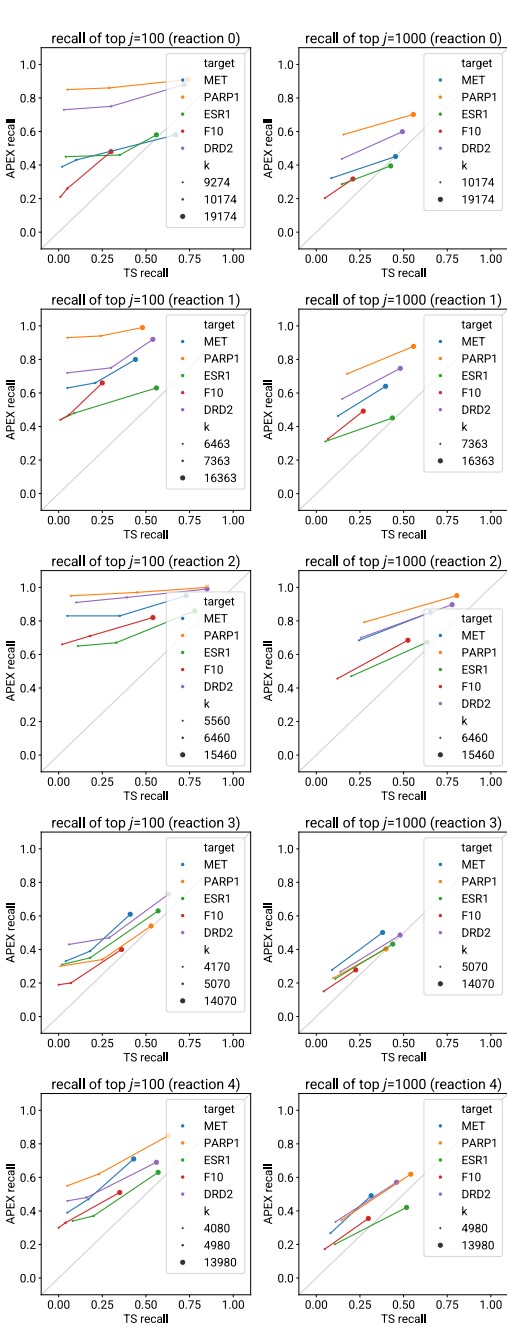

Figure 8: Top-$j$ recall for APEX and Thompson sampling (TS) using matched evaluation budgets. APEX search run using $k$ set to the number of total evaluations for TS. Thompson sampling comparison was run using three and ten warmup steps for two- and three-component reactions, respectively, and 10, 1000, or 10,000 iterations of Thompson sampling.

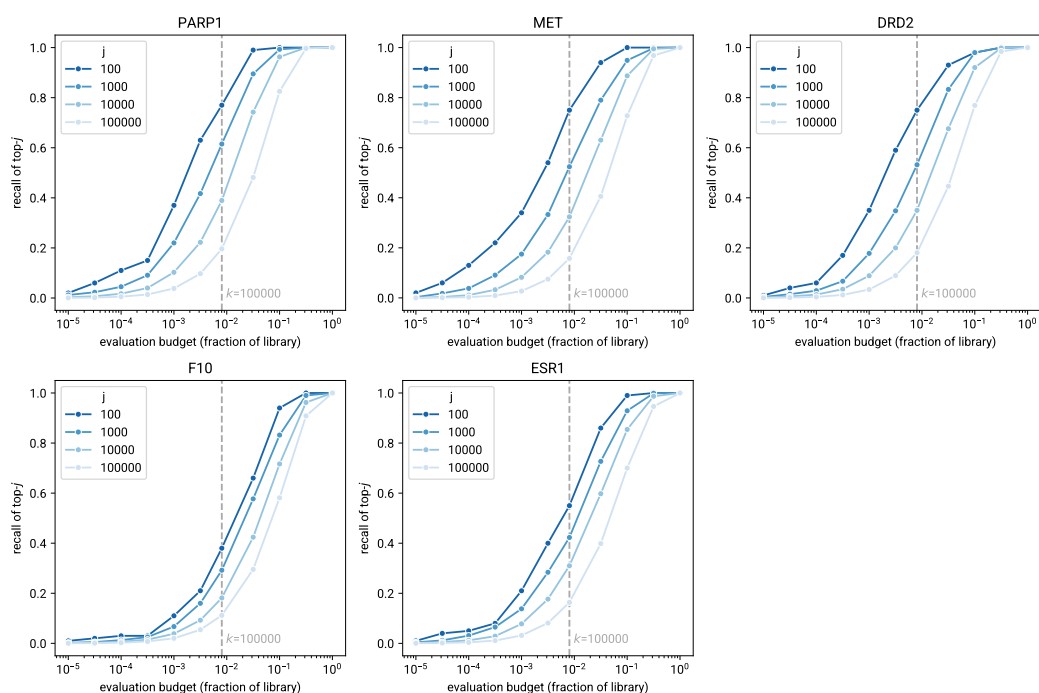

Figure 9: Recall of ground truth top-$j$ compounds at different evaluation budgets. No constraints were imposed. Dashed line corresponds to a budget of $k = 100,000$ compounds, which was used for the evaluations in Figure 3.

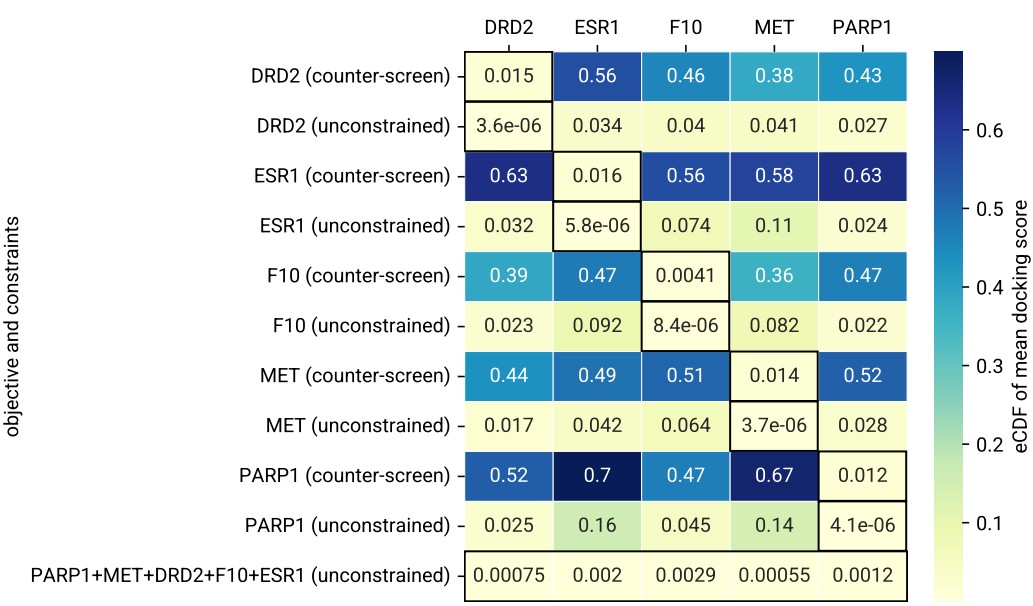

Figure 10: Inclusion of constraints on non-objective docking scores allows for APEX to be used in a counter-screening fashion. Each row is the result of a single APEX search ($k = 100,000$), either unconstrained or with "counter-screening" constraints (non-objective docking scores $>$ 50th percentile). Cells are outlined if they were used as the objective, and values are the eCDF of the mean docking score for the top 100 molecules after re-ranking by the true objective. (*Last row*) APEX search with composite objective of all five targets' docking scores.

