# OpenReview forum: "APEX: Approximate-but-exhaustive search for ultra-large combinatorial synthesis libraries"
_ICLR.cc/2026/Conference — ICLR 2026 Conference Withdrawn Submission_

### Official Review · Reviewer_Qxap · 2025-10-30

**Soundness:** 3
**Presentation:** 2
**Contribution:** 3
**Rating:** 6
**Confidence:** 2

**Summary:**

This paper presents APEX, which aims to find drug candidates from combinatorial synthesis libraries. The proposed method trains a surrogate neural network model to predict docking scores. In turn, by utilizing the hierarchical structure of CSL, the embedding of each synthon is calculated only once, and these can be combined to reconstruct the embedding of the entire compound. In this way, APEX can do an approximate-but-exhaustive search on the user's request, achieving good retrieval accuracy and runtime.

**Strengths:**

The proposed method enabled exhaustive search, which is based on approximated info, but can provide good retrieval accuracy with reasonable runtime. APEX seems to be technically sound, has clear presentation, and showed reasonable perf according to increased compound and k value.

**Weaknesses:**

1. I believe there are several existing works that presented a surrogate model; however, these are not discussed in the paper. Also, it'd be great if authors could provide a comparison to these works.
2. Can you please share how much GPU memory is consumed for training and actual inference?
3. How does associative embedding work? Can the model consider reaction in different order? e.g., when doing A+B the reaction result can be A-B or B-A, where - is bonding.
4. Is there any recent work that provides better result than Thompson Sampling?

**Questions:**

Please see Weaknesses section.

---

> ### Author Response · Authors · 2025-12-03
> **Response to reviewer Qxap [1/2]**
>
> Thank you for your review and your questions. We address the reviewer’s noted weaknesses and concerns below.
>
> **Related work and comparison**
>
> We agree that a more thorough discussion of existing surrogate based methods for virtual screening is warranted to properly contextualize the contribution of this work.
> - APEX was developed to support efficient declarative retrieval over ultra-large CSLs, such as those provided by vendors like Enamine and WuXi, where the queries involve optimizing a computational endpoint subject to computational constraints.
> - This is a common operating mode in virtual screening, where these CSLs are becoming increasingly standard due to their large size, high likelihood of synthesizability at appropriate yields, and low cost/lead time for synthesis (i.e., researchers can take the outputs of a screen on a CSL and directly order compounds from that list via the vendor).
> - Commercial CSLs have grown to a size that precludes the possibility of exhaustive labeling with computational endpoints of interest (docking or co-folding scores).
> - Surrogate based modeling is a common ingredient in a number of strategies for screening large compound libraries and work by proxying the expensive oracle with a more computationally tractable model (e.g., a neural network trained to approximate it), but even these approaches fail to scale to modern ultra-large CSLs since they would require an inordinate number of neural network evaluations per surrogate scoring iteration.
> - In this paper, we propose a design for surrogate models on CSLs that scales to the ultra-large sizes of modern day libraries and amortizes the cost of repeated screening under alternative problem formulations, making it a practical search tool for these libraries.
> - We have revised the paper to place our contribution in the appropriate context.
>
> **GPU memory consumption**
>
> All steps (surrogate training, factorizer training, and inference/search) were performed on a single NVIDIA T4 GPU, which illustrates the low resource footprint required for establishing and using an APEX model.
>
> In the GPU top-k implementation, the top-k is calculated in batches governed by a user specified chunk size. So for a batch of products (reactions, R-group assignments), their APEX predicted objective (Eq. 16) and constraint violation (Eq. 18) are materialized on the fly and entered into the priority queue. Letting C be the number of components in the largest reaction in the library, the number of floating point operations required to calculate the APEX objective and constraint violation for the entire batch is worst case (C - 1) x (m + 1) x chunk_size, where m is the number of specified constraints. For a library like the Enamine REAL library, which contains 2- and 3-component reactions (C = 3), and using a standard set of filters like the Lipinski rule-of-five constraints (m = 4), this amounts to a maximum of 10 floating point operations required to evaluate each compound. For experiments on the NVIDIA T4, we used a chunk size of 1B products, which allows us to stay under its 16GB memory limit.
>
> **How do the synthon associative embeddings work?**
>
> The factorizer is trained to approximate the molecular embedding as a sum of matrix-vector products, each of which denotes a key-value assignment, where the “key” is a matrix that depends on the (reaction, R-group) and the “value” is a vector that depends on the synthon. So permuting the assignment of synthons to R-groups (assuming such assignments are valid) will in general result in different sums of matrix-vector products (essentially it just permutes which matrices are multiplied to which vectors). The exception is for cases where the R-groups are isomorphic (i.e., contain the exact same set of synthons and the R-group position in the reaction are isomorphic), in which case the factorized embedding is invariant to such permutations.

---

> > ### Author Response · Authors · 2025-12-03
> > **Response to reviewer Qxap [2/2]**
> >
> > **Alternative baselines?**
> >
> > We acknowledge that the baselines presented in the paper are limited, as our primary focus is a methodological one on a different problem specification than what has been the typical focus in the literature on ML for virtual screening where active learning approaches are more prevalent.
> > - We believe that there is a crucial distinction between the problem APEX is addressing - namely, being able to amortize the cost of running multiple distinct searches on a given library as the design criteria for a drug discovery program mature - as opposed to methods more firmly rooted in the active learning literature, which treat each new search specification (objective + new set of constraints) as a completely distinct search problem. -- To optimize against a new target or constraint set, active learning approaches would typically require a completely new and often time consuming iterative cycle. The budgets across distinct searches therefore scale roughly linearly in the regime where each search problem is governed by a separate active learning loop.
> > APEX overcomes this through its clever amortization strategy. Once the surrogate and factorizer are trained, the expensive computational work is complete. For running inference, changing the objective or constraints can be handled at runtime, and executing such a search on a 10B compound CSL is extremely fast on a modern consumer GPU (< 1 minute).
> > - We chose the comparison to Thompson sampling due to it being a strong and established baseline and there being an open implementation that has been rigorously benchmarked and that works natively on Enamine-like CSLs. We do not know of recent work that outperforms Thompson sampling for the purpose of screening Enamine-like CSLs.
> > - We recognize that there are a number of more recent approaches in the literature that innovate on components of the active learning workflow for virtual screening, but many of these are not directly applicable to ultra-large CSLs such as those offered by Enamine and WuXi due to their high compute requirements (see response to reviewer 3n24 regarding the number of GPU hours required for surrogate evaluation in MolPAL on libraries of this size).

---

### Official Review · Reviewer_Wn7N · 2025-10-31

**Soundness:** 3
**Presentation:** 3
**Contribution:** 3
**Rating:** 4
**Confidence:** 4

**Summary:**

This paper presents APEX (Approximate-but-Exhaustive Search), which is a neural network-based framework for virtual screening of ultra-large combinatorial synthesis libraries (CSLs). The core challenge addressed is that modern CSLs contain tens of billions of make-on-demand compounds, but computational constraints typically limit virtual screening to evaluating less than 0.1% of available compounds, potentially missing many high-scoring candidates. APEX addresses this limitation through a three-step approach: (1) training a multi-task surrogate model that predicts molecular properties from molecular representations using a neural network encoder; (2) training a reaction factorizer that exploits the combinatorial structure of CSLs to reconstruct the surrogate's embeddings from reaction and R-group assignment pairs, thereby enabling efficient amortization across the entire library; and (3) performing approximate-but-exhaustive search by computing factorized surrogate predictions for all compounds in the CSL and retrieving the top-k compounds that maximize a user-specified objective subject to constraints. The key innovation is the APEX factorization, which decomposes surrogate predictions into synthon associative contributions that can be precomputed and cached, reducing the computational complexity.

**Strengths:**

1. I appreciate that the authors explicitly address real-world virtual screening constraints that are often overlooked in academic papers. The paper recognizes that modern CSLs contain tens of billions of compounds, yet computational budgets typically allow evaluation of less than 0.1% of the library, which is a genuine bottleneck in industrial drug discovery settings. The emphasis on handling multiple constraints is also valuable, as constraint satisfaction is critical in real-world drug discovery applications.

2. The runtime performance is remarkable—screening over 10 billion compounds in approximately 30 seconds on a single Tesla T4 GPU (Table 1).

3. The factorization approach that decomposes molecular embeddings into synthon associative contributions is well-explained.

**Weaknesses:**

1. The text size in Figure 1 and Figure 2 is extremely small and nearly illegible at normal viewing resolution. I had to zoom to 500% magnification to read the text labels, annotations, and diagram components. The authors should significantly increase font sizes and improve the overall figure format.

2. The paper makes claims about screening 10 billion compounds using a surrogate model trained on only 1 million, yet provides insufficient evidence to support the reliability of this generalization. The fundamental assumption is that a surrogate model trained on 1M compounds can accurately predict properties across the full 10B chemical space, yet there is no theoretical or empirical justification for this distribution shift. In real-world deployment, the model would likely encounter compounds with unseen scaffolds and functional groups, where the reliability of the surrogate remains untested and uncertain.

3. APEX introduces a two-stage approximation: first, the surrogate model approximates the true objective, and then the factorizer approximates the surrogate. However, the cumulative error arising from these cascaded approximations is neither quantified nor theoretically bounded. It remains unclear how the authors ensure that error accumulation does not occur within their proposed framework.

4. The authors inject isotropic Gaussian noise into embeddings during training and claim this makes predictions robust. However, this is merely a statistical regularization technique that assumes embedding perturbations follow a simple normal distribution. This does not reflect the actual uncertainty structure in chemical space, where small structural changes can cause large, non-random shifts in molecular properties. The noise injection is mathematically convenient but chemically unmotivated. If the authors believe this approach is chemically grounded or relevant, please provide a detailed justification and explanation.

5. The paper compares APEX against only a single baseline method (i.e., Thompson sampling). The introduction mentions multiple recent virtual screening algorithms (V-SYNTHES  and NGT) but provides no experimental comparison with any of them. If the authors consider Thompson sampling to be the only valid baseline, they should explicitly justify this choice and clarify why other established approaches were excluded from evaluation.

**Questions:**

1. The paper implements custom PyTorch CUDA C++ extension modules for top-k retrieval with chain-of-batches strategy and GPU-compatible AIR top-k algorithm. Does this custom hardware-level implementation mean that users would need to modify or fine-tune the CUDA code for different GPU architectures, library sizes, or constraint specifications? How portable is this implementation across different hardware (e.g., AMD GPUs, Apple Silicon, newer NVIDIA architectures)?

2. What is the trade-off between embedding dimension, surrogate accuracy, factorization fidelity, and memory consumption? Should users with larger synthon libraries or more complex objectives use higher d? Is there a principled way to determine the minimum d needed for a given retrieval accuracy target?

3. The paper assumes CSLs follow a specific hierarchical structure with reactions, R-groups, and synthons. How does APEX handle: (1) libraries with variable numbers of components per reaction (e.g., some reactions use 2 synthons, others use 4 or 5); (2) multi-step synthesis routes rather than single-step combinatorial reactions; (3) libraries with reaction templates that have positional dependencies (e.g., where synthon choice for R-group 1 constrains valid choices for R-group 2)?

4. All evaluations in the paper use predicted docking scores from GPU-accelerated AutoDock Vina as ground truth. However, docking scores are themselves approximations of true binding affinity, and there is often poor correlation between computational docking and experimental binding measurements. Have any of the APEX-retrieved compounds been validated experimentally?

5. As commercial CSLs continue to grow toward 100 billion or even trillion-scale libraries, how does APEX's performance scale? Does retrieval accuracy decrease as the library grows and the chemical space becomes more diverse?

6. APEX performs a single-shot retrieval based on the trained surrogate and factorizer. In practice, virtual screening is often iterative: initial hits are validated, and the model is refined based on feedback. Does APEX support active learning workflows, or is APEX intended only as a one-time screening tool, requiring full retraining if the user wants to incorporate new data?

---

> ### Author Response · Authors · 2025-12-03
> **Response to reviewer Wn7N [1/3]**
>
> Thank you for your review and your questions. We address the reviewer’s noted weaknesses and concerns below.
>
> **Figure legibility**
>
> We apologize for the poor legibility of the figures and have addressed this by increasing font sizes for labels and annotations in the illustrative Figures 1 and 2 to ensure they can be read easily at standard viewing resolutions and have improved the figure layout for better clarity.
>
> **Reliability of generalization**
>
> The reviewer raises a valid concern about the generalization of the surrogate trained on 1M compounds when inferring on a library of 10B compounds. Our justification is as follows:
> - All three CSLs (1M, 12M, and 10B) developed for the purposes of this paper are by design related up to exchangeability due to the library construction process. In real world applications, exchangeability can be similarly guaranteed according to these steps:
>   - Take the original ultra-large combinatorial library that is of interest for screening (e.g., 10B)
>   - Specify a training set size (e.g., 1M)
>   - For each reaction, sample an equal proportion of synthons within each R-group such that the relative frequencies of reactions and size of R-groups within each reaction are maintained in the new subsampled CSL as in the original.
>   - This new mini CSL is related to the original CSL up to exchangeability, hence, the expected generalization risk for a model trained on the mini CSL is equivalent to that of the original CSL, which suggests that the relationships learned by the surrogate are transferable to the 10B combinatorial space.
> - For empirical support, we refer the reviewer to Figure 7 which includes the per endpoint R-squared on the heldout 12M set.
> - As an assessment of APEX’s zero-shot generalization to a new, distinct library, we have included a new set of results that demonstrates the application of the trained surrogate and factorizer to the 9/2024 Enamine REAL library, a widely used make-on-demand vendor CSL which contains more than 70 billion compounds. Both the synthons and reaction templates in the REAL library are distinct from the BRICS CSLs. The newly updated Figure 7 reports on the R-squared and measures of rank correlation between the factorized and unfactorized surrogate predictions for the docking score endpoints on a randomly sampled subset of 100,000 compounds from the REAL library in comparison to the performance on the 1M BRICS CSL used in training. Although these measures of fitness decrease on the new library as would be expected, we note that (a) the degradation in performance is perhaps less than would be feared (largest drop is for MET, where the Spearman rho for the factorized surrogate goes from 0.61 in the BRICS 1M to 0.43 on the Enamine REAL 70B), and (b) the updated Figure 4 now includes docking score curves for the APEX top-k run on the REAL 70B library in a zero-shot fashion and shows clear enrichment in docking scores of compounds retrieved by APEX, which **demonstrates that APEX can be utilized on a new unseen library to identify high scoring compounds.**
>
> **Isotropic noise injection**
>
> The reviewer is correct that the injection of Gaussian noise in Eq. 4 is purely for purposes of statistically regularizing the predictor. This is simply to address the classical errors-in-variables problem in regression that is introduced when we substitute the original molecular embeddings with the factorized embeddings meant to approximate them (thereby introducing error into the independent variables in the regression). By training the objective heads to be robust to random, small perturbations in the embedding space, the subsequent introduction of actual noise (via substitution with the factorized embedding) is mitigated. We have clarified that this technique is a mathematical strategy to stabilize the subsequent factorization, rather than a claim about chemical uncertainty modeling (which is not its purpose).
>
> **Portability of CUDA implementation**
>
> While the core PyTorch operations and CPU implementation is portable across accelerators, the highly optimized CUDA implementation utilizes NVIDIA-specific libraries (CCCL/AIR top-k) and is not directly portable to other vendors (e.g., AMD, Apple). Supporting these architectures would require re-implementation using the respective vendor’s low level libraries. For now, APEX’s peak performance is realized on the NVIDIA ecosystem. No changes are needed to the CUDA operator to use different libraries or search constraints.
>
> **Alternative baselines?**
>
> Please see response to reviewer Qxap.

---

> > ### Author Response · Authors · 2025-12-03
> > **Response to reviewer Wn7N [2/3]**
> >
> > **Trade-offs in embedding dimension, accuracy, memory, and runtime**
> >
> > The embedding dimension d is the key parameter dictating the trade-offs in accuracy vs. memory consumption, but other factors like the encoder architecture are certainly relevant as well. However, since our primary contribution is in how the embeddings are used and factorized to accelerate full inference over a CSL, let’s focus on the embedding dimension and not the encoder (as this is in some sense an arbitrary choice).
> > - Accuracy: Since the surrogate is trained for multi-task prediction, the embedding dimension d upper bounds the rank of the predicted outcome matrix and thereby imposes a natural limit on the average of per-endpoint prediction accuracy.
> > - Memory: The memory consumption is dominated by storing the synthon associative contributions, which is of dimension $m$ (the number of tasks), not $d$. Larger embedding dimension will increase the memory and runtime for this pre-computation/amortization step, but does not change the storage requirements.
> > - Runtime: The APEX runtime is not a function of $d$: once the contributions are pre-computed, the final step just requires $O(|X_\mathcal{D}|)$ summations regardless of $d$.
> >
> > **Handling of different CSL structures**
> >
> > APEX can handle common CSL variations (in some cases, it will require generalizations to the implementation, but can be done), but does have limits:
> > 1. **Libraries with variable number of components per reaction:** APEX natively handles this. The final factorized prediction is a summation over all R-group and synthon assignments, and is agnostic to the number of components in a reaction. The CSLs that we are including for purposes of evaluation include multiple 2- and 3-component reactions (like the Enamine REAL library).
> > 2. **Multi-step synthesis:** The current design and exposition in this paper is focused on single-step combinatorial reactions. We note that it is possible to represent multi-step reactions as multiple single-step reactions, which is commonly done in commercial CSLs, but recognize this can have limitations. Modeling multi-step reactions natively would require a more complex factorization strategy and is outside the scope of the current method.
> > 3. **Positional dependencies:** This is something that is not supported in the current implementation, but can certainly be accommodated. If the validity of an R-group assignment can be determined as an O(1) lookup, we could simply check at runtime first if an assignment is valid and skip cases that are not (i.e., we would not sum the synthon associative contributions for invalid cases and they would be ineligible for the priority queue).
> >
> > **Experimental validation of APEX-retrieved compounds**
> >
> > - We acknowledge that docking scores are just computational approximations of binding affinity. Our contribution is a methodological one, namely the development of an efficient and scalable search protocol for ultra-large CSLs against computational oracles. We apply it to the problem of optimizing docking scores and demonstrate that APEX successfully recalls a high proportion of top scoring compounds.
> > - Experimental validation of APEX-retrieved compounds was not performed for purposes of this study, but we are actively pursuing such validation and intend to include these results in future studies.
> >
> > **Scaling to larger CSLs**
> >
> > The primary computational step is the summation in Eq. 16, which requires $O(C\cdot|X_\mathcal{D}|)$ summations, where $C$ is the number of reaction components (typically 2 or 3). Hence, APEX’s performance scales linearly with the number of compounds in the CSL.
> > - Screening a 1T compound library would require ~100x more compute time than the ~30 seconds reported for the 10B compound CSL we used for purposes of evaluation. Extrapolating this linearly suggests a runtime of roughly 50 minutes on a single Tesla T4 GPU, which is still many orders of magnitude faster than conventional methods.
> > - We have updated Table 1 to include the runtime for APEX top-k searches on the 09/2024 Enamine REAL library, which contains 70B compounds, and updated the score distributions in Figure 4 accordingly which highlights the enrichment in docking scores attained from screening the REAL library. As reported in Table 1, screening on the 70B REAL library took roughly three minutes.
> > - The reviewer raises an important question on the relationship between retrieval accuracy and library size. For a fixed k, as the library size grows, the potential for encountering high scoring compounds that are not in the ground truth top-k set increases. However, constructing a large, fully enumerated and labeled set for libraries of such size is a limitation to carrying out a thorough analysis. Instead, we can assess the extent to which the computational oracle (docking scores in this case) are enriched for the APEX top-k set as the size of the library grows. We have included additional curves in Figure 4 to illustrate this.

---

> > > ### Author Response · Authors · 2025-12-03
> > > **Response to reviewer Wn7N [3/3]**
> > >
> > > **Does APEX support active learning or iterative refinement to its search?**
> > >
> > > APEX was designed with the goal of facilitating fast, declarative, single-shot searches on ultra-large CSLs. We believe that an active learning variant of APEX would be a valuable extension, but feel that developing and benchmarking such a workflow would constitute a separate, significant research effort and falls outside the methodological scope of the current paper.
> > > - We should note, however, that in the current design, new labeled data (either new compounds labeled with existing endpoints or existing compounds labeled with new endpoints) can be used to fine-tune the surrogate and subsequently fine-tune the factorizer so that APEX searches can be run on models effectively updated on new data.

---

### Official Review · Reviewer_5r6H · 2025-11-01

**Soundness:** 2
**Presentation:** 1
**Contribution:** 1
**Rating:** 2
**Confidence:** 3

**Summary:**

This manuscript introduces APEX (Approximate-but-Exhaustive), a search protocol for ultra-large combinatorial synthesis libraries (CSLs) that aims to make “full-library” virtual screening feasible by combining (i) a neural surrogate trained on an enumerated subset and (ii) a factorization mechanism that decomposes surrogate embeddings into synthon-level contributions. This enables the model to evaluate all compounds implied by the CSL (up to 10B+ products) on a single GPU and then perform top-k retrieval under user-specified objectives and constraints.

**Strengths:**

- Problem importance: The paper tackles a very real bottleneck in structure-based and library-based discovery: current make-on-demand CSLs are now so large (10⁹–10¹⁰) that even “smart” virtual screening methods end up seeing <1% of the space, leaving high-scoring chemistry unexplored. A method that makes declarative, low-latency queries over the entire library is valuable to both method developers and practitioners. The paper also recognizes that constraints (Lipinski, Veber, fragment-like rules, etc.) are not an afterthought but central to realistic screening campaigns.

- Interesting and novel factoring mechanism: The key technical idea of learning a factorizer that reconstructs the surrogate embedding from (reaction, R-group, synthon) choices, is genuinely interesting. By pushing the complexity into a hierarchical, library-aware model (synthon → R-group → reaction → associative contributions), APEX can precompute a small number of neural evaluations and then reuse them combinatorially when traversing the CSL. This is a nice instantiation of “amortization over a structured chemical space,” and the linear projection to task heads (eqs. (13)–(16)) makes the top-k scan GPU-friendly. It is a nontrivial step beyond simply “batch the surrogate on GPU.”

**Weaknesses:**

- Dependence on synthon + reaction paths for every library compound: APEX only works because every product in the library is addressable as “reaction + R-group + synthon” and the factorizer has been trained on that exact CSL structure. This is fine for Enamine-style, synthon-organized libraries, but it also means the method is not directly applicable to an arbitrary compound library (e.g., a corporate merged screening collection, ChEMBL-like flat sets, or ad-hoc AI-generated enumerations) unless that library can first be expressed in this hierarchical CSL form. This limitation should be made explicit in the paper: APEX solves the CSL case, not the general VS case. Right now the narrative leans toward “fast virtual screening on massive libraries,” but the technical requirement is actually “fast screening on massive combinatorial libraries with known factorization.” Please clarify scope and portability, and discuss what would be needed to support (i) multiple CSLs with different factorization schemes and/or (ii) partially specified synthons.

- Benchmarked properties are quite basic: The experimental section mostly shows APEX recovering top docking hits and satisfying standard medicinal chemistry filters (Lipinski, Veber, Astex Rule of 3, Pfizer 3/75). These are useful sanity checks but they are all relatively low-level or RDKit-level properties. For a method whose selling point is “once your surrogate is trained, you can query anything,” it would be much more convincing to see: docking-like tasks with tighter resolution (e.g., distinguishing top 0.01% vs. top 0.1%); multi-target or composite objectives; a direct comparison on docking-score enrichment against baselines.

- Right now, the tasks mostly show that the factorization is not breaking the surrogate, but they don’t fully stress-test how well APEX preserves ordering on a hard, high-variance objective like docking. The paper would be much stronger with more analysis specifically on docking-score tasks, including per-target recall@k curves and order-preservation (Kendall-τ / Spearman-ρ) between surrogate, factorized surrogate, and ground truth.

**Questions:**

N/A

---

> ### Author Response · Authors · 2025-12-03
> **Response to reviewer 5r6H**
>
> Thank you for your review and your questions. We address the reviewer’s noted weaknesses and concerns below.
>
> **Dependence on the CSL structure and clarification of scope**
>
> We of course agree that this limitation should be made explicit and have clarified the technical scope of APEX within the revised manuscript. To be clear:
> - The core function of APEX is predicated on the known combinatorial structure of a CSL. Specifically, that every molecule x can be represented by a multi-index $\xi = (t,\{(r,s)\})$ defining the rection ($t$), R-group assignments ($r$), and synthons ($s$). This structure is essential because it is what enables the factorization and amortization of the surrogate model. We explicitly clarify that **APEX solves the specific problem of fast, exhaustive search on ultra-large CSLs with a known factorization structure**, and is not intended for general unstructured molecular libraries. It is worth noting that many of the most commonly used make-on-demand CSLs, such as those offered by Enamine and WuXi, admit this structure by design.
>
> Regarding APEX’s portability:
> - APEX was not designed to work on arbitrary enumerated compound libraries like ChEMBL; it is designed for libraries that admit combinatorial structure.
> - APEX was specifically designed for CSLs in which all products can be expressed as a reaction and R-group assignment. This is the most common design for commercially available make-on-demand combinatorial libraries, which is the focus of our work. Although the key ideas we exploit in the factorization can likely be extended to alternative combinatorial library designs, it is outside the scope of the current work.
>
> **Benchmarked properties and constraint sets are too basic**
>
> We acknowledge the reviewer’s request for more challenging constraint sets compared to those considered in the paper. We have revised the paper to include additional, relevant, and low-prevalence constraint sets such as those targeting CNS penetration, as well as experiments on the more challenging problem of counter-screening (optimizing the docking score of one target subject to a constraint on another target’s docking score) to further demonstrate APEX’s capability as a search tool for ultra-large CSLs.
> - The newly included Figure 10 applies APEX to a counter-screening task in which the goal is to minimize the docking score for a given target subject to the constraint that the docking score on the alternative targets exceeds the 50th percentile value (evaluated on the background distribution). For each query, we run an APEX top-k search with k=100,000 on the 12M BRICS CSL and select the top 100 from this after re-ranking by the actual (oracle) docking score objective. For these 100 compounds, we calculate the mean docking score across each of the five targets (the one objective and four used as constraints) and evaluate the empirical CDF of this mean value. For all targets, we clearly see that the compounds identified by APEX in the presence of the counter-screening constraints attain low docking scores for the target and high docking scores for the off-targets, **demonstrating APEX’s applicability to the more challenging and realistic setting of counter-screening.**
> - Per the reviewer’s request for more comprehensive recall-j-at-k curves, we have included a new such panel in Figure 3C (aggregated) as well as a more extensive version in the newly added Figure 9 (broken out by target) in the supplement.

---

### Official Review · Reviewer_3n24 · 2025-11-01

**Soundness:** 2
**Presentation:** 1
**Contribution:** 1
**Rating:** 2
**Confidence:** 4

**Summary:**

This paper proposes APEX, a surrogate-based  search framework for ultra-large combinatorial synthesis libraries. The workflow trains a surrogate model to predict an objective (docking), then trains a reaction factorizer that reconstructs the surrogate embedding from reaction, R-group, and synthon components. The model then performs GPU-accelerated top-k retrieval over the entire enumerated CSL using pre-computed, factorized contributions. The authors evaluate APEX on docking tasks for five targets and claim quickly retrieval for 1 M–12 M-sized datasets, extending to "potential" 10 B-scale enumeration.

**Strengths:**

1. Evaluation of the method is reasonable:  Includes five realistic molecular targets and measures runtime and recall, showing consistent acceleration over brute-force screening.

**Weaknesses:**

1. Exceptionally limited related work. The paper omits extensive prior work in active learning and surrogate-based virtual screening, such as
    - MEMES: Machine Learning Framework for Enhanced Molecular Screening [1]
    - Accelerating High-Throughput Virtual Screening through Molecular Pool-Based Active Learning [2]
    - Generative AI for Navigating Synthesizable Chemical Space [3]

These works already report more sophisticated active learning and synthesizability-aware loops, which APEX neither cites nor compares against. Before publication, the paper requires a thorough literature review of enumeration and active learning in molecular design to contextualize its contribution.

2. Novelty is weak. Using a learned surrogate for pre-screening enumerated chemical spaces is well-established. APEX mainly implements an efficient factorization of embeddings rather than a new search paradigm.
3. The “open CSL” is created by without synthesizabilty consdierations, so generated compounds are not necessarily synthesizable, unlike Gao et al.’s synthesizability approaches [3]
4. Baseline selection is poor. All comparisons are against non-ML heuristics (random search, Thompson sampling) rather than established machine learning-based or active learning approaches

**Questions:**

1. Definition of “approximate-but-exhaustive”: Please formalize what is meant by “exhaustive search” when using a factorized surrogate. Is there any theoretical guarantee that top-k ranking under the surrogate approximates the true top-k under the physical objective?
2. Baselines: Why are active learning and ML-based selection strategies excluded from comparison? How would APEX perform relative to these?
3. Could the authors contextualize the work within the broad landscape of chemical search using ML ?

---

> ### Author Response · Authors · 2025-12-03
> **Response to reviewer 3n24 [1/2]**
>
> Thank you for your review and your questions. We address the reviewer’s noted weaknesses and questions below.
>
> **Limited related work and missing baselines**
>
> We recognize the reviewer’s critique on the lack of comparisons against prior work on active learning, surrogate-based virtual screening, and generative models over synthesizable chemical space. We acknowledge the distinction and stress that the critical challenge APEX addresses is the ability to effectively enumerate through ultra-large CSLs with neural network surrogates, which fundamentally separates our method from prior work.
> - Active learning (e.g., [1] and [2]) and generative methods (e.g., [3]) operate under a limited, sequential evaluation budget for optimizing a costly ground truth function with the goal of maximizing the expected reward under a limited number of evaluations. In contrast, APEX is designed for rapid retrieval of declarative queries on ultra large combinatorial libraries by facilitating exhaustive evaluation on these search spaces.
> - It is correct that APEX is a surrogate based approach for virtual screening, and that methods like MolPAL too rely on scoring a large pool of candidate molecules using a surrogate model (e.g., a GNN/transformer or GP) to select the next batch of molecules for ground truth evaluation. We acknowledge that these active learning counterparts place a greater emphasis on the active learning part of the workflow, which has not been a focus of the APEX work.
> - Modern make-on-demand CSLs, such as those offered by vendors like Enamine and WuXi, number in the tens of billions of compounds. **For libraries of this size, evaluating even a highly efficient GNN/transformer/GP surrogate model across the entirety of the search space is computationally infeasible, which is the problem we are specifically addressing with APEX.**
> - APEX’s amortized solution is unique and overcomes this computational barrier by exploiting the combinatorial structure of CSLs to construct efficient approximations. We demonstrate this empirically by showing retrieval results on a 10B+ compound CSL in under 30 seconds using a single NVIDIA Tesla T4 GPU.
> - In contrast, the MolPAL GitHub lists [timing results](https://github.com/coleygroup/molpal?tab=readme-ov-file#timing) where MPN prediction on 100M molecules required 4 x V100s for 2 hours. Naively scaling this to a 10B compound library would require 4 x V100s for 200 hours (i.e., 800 GPU hours) for each iteration of active learning. **Hence, standard surrogate based methods like MolPAL that do not exploit the combinatorial structure of the search space are unable to scale to ultra-large CSLs that are of interest in hit discovery campaigns.**
>
> **Novelty**
>
> We agree that using a learned surrogate for pre-screening is established. However, we contend that our strategy for accelerating exhaustive inference via the designed factorization of the surrogate embedding is novel and impactful in enabling efficient screening of ultra-large CSLs.
> - The factorization converts exhaustive surrogate screening into a problem that requires $O(|X_\mathcal{D}|)$ full neural network evaluations into an amortized summation of pre-computed terms that amounts to $O(|X_\mathcal{D}|)$ floating point operations, which can be carried out efficiently on a modern GPU.
> - This specific combinatorial factorization is what enables exhaustive surrogate evaluation on 10B+ compound libraries in under a minute with just a single T4 GPU.
>
> **Definition of “approximate-but-exhaustive”**
>
> The term “approximate-but-exhaustive” refers to exhaustive search over the space using the factorized surrogate predictions (which is an approximation of the unfactorized surrogate predictions).
> - Approximate: The original surrogate model is approximated by the factorized surrogate, which substitutes the molecular embeddings with the associative form (Eq. 11).
> - Exhaustive: The search is exhaustive in that every single compound $x\in X_\mathcal{D}$ in the CSLis evaluated using the factorized surrogate (Eq. 16).
>
> Regarding theoretical guarantees:
> - The factorized surrogate is trained to minimize reconstruction error of the factorized embedding (Eq. 11) with respect to the true surrogate embedding (Eq. 12).
> - The original surrogate is trained to minimize the prediction error with respect to the ground truth property values (Eq. 3).
> - This two-step process does not guarantee that the top-k ranking under the factorized surrogate is the true top-k ranking under the physical objective. However, we empirically demonstrate its effectiveness via high recall metrics (Figure 3A), which measures the overlap between the APEX top-k set and the ground truth top-k set, confirming that APEX offers an effective approximation to the true top-k set.

---

> > ### Author Response · Authors · 2025-12-03
> > **Response to reviewer 3n24 [2/2]**
> >
> > **Synthesizability of the open CSL**
> >
> > We appreciate the reviewer’s concern regarding the synthesizability of the open CSLs developed for the purposes of this paper. We would like to clarify the object behind the creation of these libraries:
> > - The open CSLs were designed for methodological benchmarking and public reproduction purposes. Our goal was to provide a large, annotated dataset (over 10 million compounds with ground-truth docking scores and physiochemical properties) that is not subject to property vendor or licensing restrictions so as to enable transparent benchmarking for screening CSLs.
> > - Our focus was not on explicitly curating the open CSLs so as to guarantee synthesizability, but to admit the same data structure as libraries provided by vendors like Enamine and WuXi (i.e., synthons and single step reactions), which are becoming ubiquitous in hit discovery efforts.
> > - **APEX is explicitly designed to operate on real-world make-on-demand CSLs** such as the Enamine REAL library, which is renowned for containing billions of compounds with a high degree of guaranteed synthetic feasibility.
> > - To demonstrate APEX’s direct applicability and performance on real-world, ultra-large CSLs that are designed with synthesizability in mind, **we have included additional results reporting the runtime on the 70B+ compound Enamine REAL library** (09-2024 version) in an updated Table 1. Furthermore, we updated the ECDF plots in Figure 4 to show the docking score distributions for the APEX top-k compounds retrieved from the REAL library in a zero-shot fashion (i.e., no further training of the surrogate or factorizer), which **clearly demonstrates enrichment in high-scoring compounds on realistic, ultra-large, make-on-demand CSLs.**

---

### Note · Authors · 2026-01-26

I have read and agree with the venue's withdrawal policy on behalf of myself and my co-authors.

---

### Meta-Review · Area_Chair_b1fh · 2026-01-06

**Summary:**

Overall, the reviewers have expressed concerns limited methodological innovation contributed by this manuscript, insufficient comparison against other relevant existing methods as baselines, applicability and usefulness of the proposed method in a general setting.
The authors attempted to provide extensive point-by-point response to the reviewers concerns, which has clarified the main focus and setting of the proposed method and the reasons underlying the authors chosen baselines for comparison.

While the authors response have made several important clarifications (e.g., the main focus of the proposed method and the primary setting of interest), the rebuttal doesn't seem to directly address the fundamental concerns regarding novelty, generalizability, and comparison against other existing ML-based virtual screening approaches.

**Reviewer Concerns:**

The authors have made clarifications regarding the main purpose of the proposed screening method, its novelty, and the primary application setting of interest, which helps understanding the primary focus of the current paper.

The authors argue that the existing ML approaches are not relevant to the proposed APEX, hence the lack of comparison (except for comparison to Thompson sampling), I tend to agree with the reviewers that the authors should make further efforts to compare APEX with other virtual screening approaches previously proposed in the field to convincingly demonstrate the advantages and uniqueness of APEX.

**Reviewer Scores:**

For reviewer 3n24, the authors rebuttal has clarified his/her doubts about what is meant by "approximate-but-exhaustive". But the primary concerns regarding the highly limited comparison to other broadly related schemes are still outstanding and lack of comparison against other ML baselines also remain.
As a result, it is unlikely that the reviewer may have increased the score substantially from the current rating of 2.

Similarly, reviewer 5r6H had concerns about the general applicability and utility of the propose APEX and concerns about the simplicity of the benchmarked properties.
In response, the authors have provided additional results, but the added results do not seem to be sufficient to demonstrate APEX utility in complex cases suggested by the reviewer (such as docking tasks with very high precision or multi-target docking, etc.)
For this reason, it is unlikely that the reviewer may have significantly increased the score from the current rating of 2.

For reviewer Wn7N, the authors have addressed some minor concerns regarding figure legends, statistical regularization through injection of Gaussian noise, etc.
However, additional comparison requested by the reviewer hasn't been sufficiently addressed, the review score may have been only slightly increased from the 4 (if increased at all).

Finally, reviewer Qxap gave the highest score of 6, but the review comments were fairly generic and light, and it's unclear whether the score may have improved further due to the rebuttal.

---

### Decision · Program_Chairs · 2026-01-26

Reject